# UBL3 modification influences protein sorting to small extracellular vesicles

Hiroshi Ageta[1,2], Natsumi Ageta-Ishihara[3], Keisuke Hitachi [1], Ozge Karayel[4], Takanori Onouchi[5], Hisateru Yamaguchi[6], Tomoaki Kahyo[2,7], Ken Hatanaka[2,16], Koji Ikegami[2,8,18,19], Yusuke Yoshioka[9,10], Kenji Nakamura[2,17], Nobuyoshi Kosaka[9,10], Masashi Nakatani[1], Akiyoshi Uezumi[1], Tomihiko Ide[11], Yutaka Tsutsumi[5], Haruhiko Sugimura [7], Makoto Kinoshita [3], Takahiro Ochiya[9,12], Matthias Mann [4], Mitsutoshi Setou[2,8,13,14,15] & Kunihiro Tsuchida[1]

Exosomes, a type of small extracellular vesicles (sEVs), derived from multivesicular bodies (MVBs), mediate cell-to-cell communication by transporting proteins, mRNAs, and miRNAs. However, the molecular mechanism by which proteins are sorted to sEVs is not fully understood. Here, we report that ubiquitin-like 3 (UBL3)/membrane-anchored Ub-fold protein (MUB) acts as a posttranslational modification (PTM) factor that regulates protein sorting to sEVs. We find that UBL3 modification is indispensable for sorting of UBL3 to MVBs and sEVs. We also observe a 60% reduction of total protein levels in sEVs purified from *Ubl3*-knockout mice compared with those from wild-type mice. By performing proteomics analysis, we find 1241 UBL3-interacting proteins, including Ras. We also show that UBL3 directly modifies Ras and oncogenic RasG12V mutant, and that UBL3 expression enhances sorting of RasG12V to sEVs via UBL3 modification. Collectively, these results indicate that PTM by UBL3 influences the sorting of proteins to sEVs.

[1] Division for Therapies Against Intractable Diseases, Institute for Comprehensive Medical Science, Fujita Health University, 1-98 Dengakugakubo, Kutsukake-cho, Toyoake, Aichi 470-1192, Japan. [2] Mitsubishi Kagaku Institute of Life Sciences (MITILS), 11 Minamiooya, Tokyo 194-8511, Japan. [3] Division of Biological Sciences, Department of Molecular Biology, Nagoya University Graduate School of Science, Nagoya 464-8602, Japan. [4] Department of Proteomics and Signal Transduction, Max Planck Institute of Biochemistry, Martinsried 82152, Germany. [5] Department of Pathology, Fujita Health University School of Medicine, 1-98 Dengakugakubo, Kutsukake-cho, Toyoake, Aichi 470-1192, Japan. [6] Division of Biomedical Polymer Science, Institute for Comprehensive Medical Science, Fujita Health University, 1-98 Dengakugakubo, Kutsukake-cho, Toyoake, Aichi 470-1192, Japan. [7] Department of Tumor Pathology, Hamamatsu University School of Medicine, 1-20-1 Handayama, Hamamatsu, Shizuoka 431-3192, Japan. [8] Department of Cellular and Molecular Anatomy and International Mass Imaging Center, Hamamatsu University School of Medicine, 1-20-1 Handayama, Higashi-ku, Hamamatsu, Shizuoka 431-3192, Japan. [9] Division of Molecular and Cellular Medicine, National Cancer Center Research Institute, 5-1-1, Tsukiji, Chuo-ku, Tokyo 104-0045, Japan. [10] Department of Translational Research for Extracellular Vesicles, Tokyo Medical University, 6-7-1 Nishi-Shinjuku, Shinjuku-ku, Tokyo 160-0023, Japan. [11] Laboratory of Electron Microscopy, Fujita Health University Joint research support promotion facility, 1-98 Dengakugakubo, Kutsukake-cho, Toyoake, Aichi 470-1192, Japan. [12] Institute of Medical Science, Tokyo Medical University, 6-7-1 Nishi-Shinjuku, Shinjuku-ku, Tokyo 160-0023, Japan. [13] Preeminent Medical Photonics, Education & Research Center, 1-20-1 Handayama, Higashi-ku, Hamamatsu, Shizuoka 431-3192, Japan. [14] Department of Anatomy, The University of Hong Kong, 6/F, William MW Mong Block 21 Sassoon Road, Pokfulam, Hong Kong, SAR, China. [15] Department of Anatomy II and Cell Biology and Anatomy, Fujita Health University School of Medicine, 1-98 Dengakugakubo, Kutsukake-cho, Toyoake, Aichi 470-1192, Japan. [16] Present address: Neurology Tsukuba Research Department, Discovery, Medicine Creation, Neurology Business Group, Eisai Co., Ltd., 1-3 Tokodai 5-Chome, Tsukuba, Ibaraki 300-2635, Japan. [17] Present address: Nitobebunka College, Department of Clinical Laboratory Sciences, 43-16 Nakano 3-Chome, Nakanoku, Tokyo 164-0001, Japan. [18] Present address: Department of Anatomy and Developmental Biology, Graduate School of Biomedical and Health Sciences, Hiroshima University, 1-2-3 Kasumi, Minami-ku, Hiroshima, Hiroshima 734-8553, Japan. [19] Present address: JST, PRESTO, 4-1-8 Honcho, Kawaguchi, Saitama 332-0012, Japan. Correspondence and requests for materials should be addressed to M.S. (email: setou@hama-med.ac.jp) or to K.T. (email: tsuchida@fujita-hu.ac.jp)

sEVs are nanometre-sized vesicles secreted from various cell types[1]. Exosomes, a type of sEVs, derived from multi-vesicular bodies (MVBs)[2,3], mediate cell-to-cell communication by transporting proteins, mRNAs, and miRNAs[3,4]. The delivery of proteins between cells by sEVs, including exosomes, is related to tumour progression and neurodegenerative diseases[5,6]. Furthermore, neurodegenerative disease-related proteins such as amyloid beta, tau, α-synuclein, and prions are also packaged inside sEVs, and spread in the brain[6–10]. These results indicate that transportation of the intracellular proteins via sEVs contributes to various types of disease. However, the molecular mechanism by which proteins are sorted to sEVs is not fully understood.

After synthesis, proteins undergo various posttranslational modifications (PTM) that influence a variety of cellular processes. The ubiquitin-dependent modification system, one of the protein degradation systems within the cell, is involved in a variety of cellular processes[11]. We previously found that the level and localisation of the synaptic protein Vesl-1S are controlled by the ubiquitin–proteasome systems[12,13]. Furthermore, using bioinformatics analysis, we identified the SCRAPPER protein as a synaptic E3 ubiquitin ligase[14]. Several proteins are reported to have a ubiquitin-like sequence, which is referred to as a "UBL domain"[15,16]. Some UBLs have been reported to act as post-translational modifiers[15]; these include the small ubiquitin-like modifiers (SUMO)[17] and Nedd8[18]. On the other hand, as one of conserved UBLs, UBL3/MUB protein has been identifed in *Arabidopsis thaliana* and is a membrane protein localised by prenylation[19]. However, the role of UBL3/MUB as a PTM factor remains poorly understood.

In this study, we provide evidence that UBL3/MUB acts as a PTM factor that regulates protein sorting to sEVs.

## Results

**Analysis of UBL3 as a post-translational modification factor**. In the present work, to identify PTM factors, we used a bioinformatics method[14,16]. We extracted UBL3/MUB (Supplementary Fig. 1), which is known to contain a ubiquitin-like (UBL) domain and is an evolutionarily conserved membrane protein localised by prenylation in animals, filamentous fungi, and plants[19]. However, the role of UBL3 as a PTM factor remains poorly understood. To clarify the role of UBL3 as a PTM factor, we expressed Flag-UBL3 in MDA-MB-231 breast cancer cells and purified UBL3 proteins by immunoprecipitation, followed by western blotting with UBL3 antiserum. The estimated molecular weight of Flag-UBL3 is 16 kDa. Interestingly, in the Flag-UBL3 expressing cells, the UBL3 signal was observed as a smear band up to a high molecular weight only under non-reducing conditions. The smear signal disappeared after the addition of 2-mercaptoethanol (βME+) to the samples, before loading onto an SDS-polyacrylamide gel (Fig. 1a, right panel). To verify whether UBL3 modification occurs in vivo, we established *Ubl3* knockout (KO) mice and analysed PTM in brain lysates, as the expression of UBL3 in the brain is relatively high (Supplementary Fig. 2a–d). We observed endogenous UBL3-related PTM in lysates from the cerebral cortex, and showed that the degree of PTM was reduced in *Ubl3* KO mice (Fig. 1b). We next investigated UBL3 modification using UBL3 mutants; we focused on all cysteine residues as UBL3 modification is dependent on non-reducing conditions. UBL3 has only two cysteine residues at its C-terminus (C113 and C114); therefore we constructed three UBL3 mutants—C113A, C114A, and C113/114A (Fig. 1c)—and studied these along with the wild-type UBL3 (Fig. 1c–e). In the UBL3C113/114A mutant, UBL3 modification was completely abolished not only in MDA-MB-231 cells but also in every cell line examined (Fig. 1d, Supplementary

Fig. 3a). However, UBL3 modification was reduced but still retained by the UBL3C113A and UBL3C114A mutants in several cells. A CAAX motif (C, cysteine; A, aliphatic amino acid; X, any amino acid) at the C-terminus is frequently found in membrane-localised proteins[20]. The UBL3/MUB also has a CAAX motif at its C-terminus (Fig. 1c); it has been shown to be prenylated for its anchoring to membranes via C114[19]. The majority of wild-type UBL3 protein, but not the mutants, was selectively found in the membrane fraction (Fig. 1e, Supplementary Fig. 3b).

In order to study the relationship between the presence of UBL3 in the membrane fraction and UBL3 modification, we attempted to identify UBL3 mutants that were still retained in the membrane fraction despite the loss of PTM activity. Intriguingly, UBL3Δ1 which lacked only one C-terminal amino acid, was found to have lost the UBL3 modification (Fig. 1f, g, Supplementary Fig. 3a). However, UBL3Δ1 was retained in the membrane fraction, whereas UBL3Δ2 and UBL3Δ3 were not (Fig. 1h, Supplementary Fig. 3b). These data indicated that UBL3 modification was not necessary for the presence in the membrane fraction.

**UBL3 modification is indispensable for sorting of UBL3 to MVBs and sEVs**. We next examined subcellular localisation of UBL3 by immunocytochemistry, as the degree of UBL3 modification was found to be reduced in the cytoplasmic fraction (Supplementary Fig. 4a). Following the introduction of EGFP-UBL3, colocalisation with various organelles was studied. More than 40% of the total EGFP-UBL3 signal was found to be colocalised with endogenous CD63, a marker for MVBs, supporting that UBL3 is enriched in MVBs (Fig. 2a, b). UBL3 did not show colocalisation with the markers for recycling endosome (Rab11), mitochondria (COXIV), endoplasmic reticulum (Calnexin), Golgi (GM130), peroxisome (PMP70), or nuclear membrane (Lamin B1) (Fig. 2a, Supplementary Fig. 4b). Colocalisation with the early endosome marker (EEA1) or lysosome marker was detected; however, this was weak compared with the degree of colocalisation with the MVB marker (Fig. 2a, b). Then, to elucidate the relationship between UBL3 modification and localisation in MVBs, we examined colocalisation of UBL3C113A, UBL3C114A, UBL3C113/114A, and UBL3Δ1 with endogenous CD63 (Fig. 2c). Unlike wild-type UBL3, the mutants did not show colocalisation with CD63 (median values: UBL3, UBL3C113A, UBL3C114A, UBL3C113/114A, UBL3Δ1 = 41.5%, 3.0%, 2.7%, 1.9%, 2.3%, respectively). To clarify the localisation of UBL3 in MVBs and the plasma membrane, we performed immunoelectron microscopic analysis (Fig. 2d, e). When Flag-UBL3 was introduced to MDA-MB-231 cells, the UBL3 signal was clearly detected in the MVBs and the plasma membrane, but not in mitochondria or the nuclear membrane (Supplementary Fig. 4c, d). UBL3Δ1 was localised in the plasma membrane but not in MVBs (Fig. 2d, e). These results indicate that membrane localisation is not sufficient and that UBL3 modification is required for MVB localisation. One fate of MVBs is to fuse with the plasma membrane for release as exosomes. The other fate is to fuse with lysosomes, which degrade the MVB contents[2,3]. In the present study, we focused on the extracellular secretion of UBL3, as this protein colocalises with MVBs to a degree greater than with lysosomes (Fig. 2a, b). Cells release extracellular vesicles (EVs) of various sizes. These vesicles, some of which are derived from the plasma membrane, include large- and medium-sized EVs purified at 2000 x *g* (2K) and 10,000 x *g* (10K), respectively. sEVs are purified at 100,000 x *g* (100K) and are derived in part from the plasma membrane (non-exosomal sEVs) or are derived from MVBs (exosomes)[21]. We have purified EVs from 2K-, 10K-, and 100K-centrifuged pellets from MDA-MB-231 cells and found that

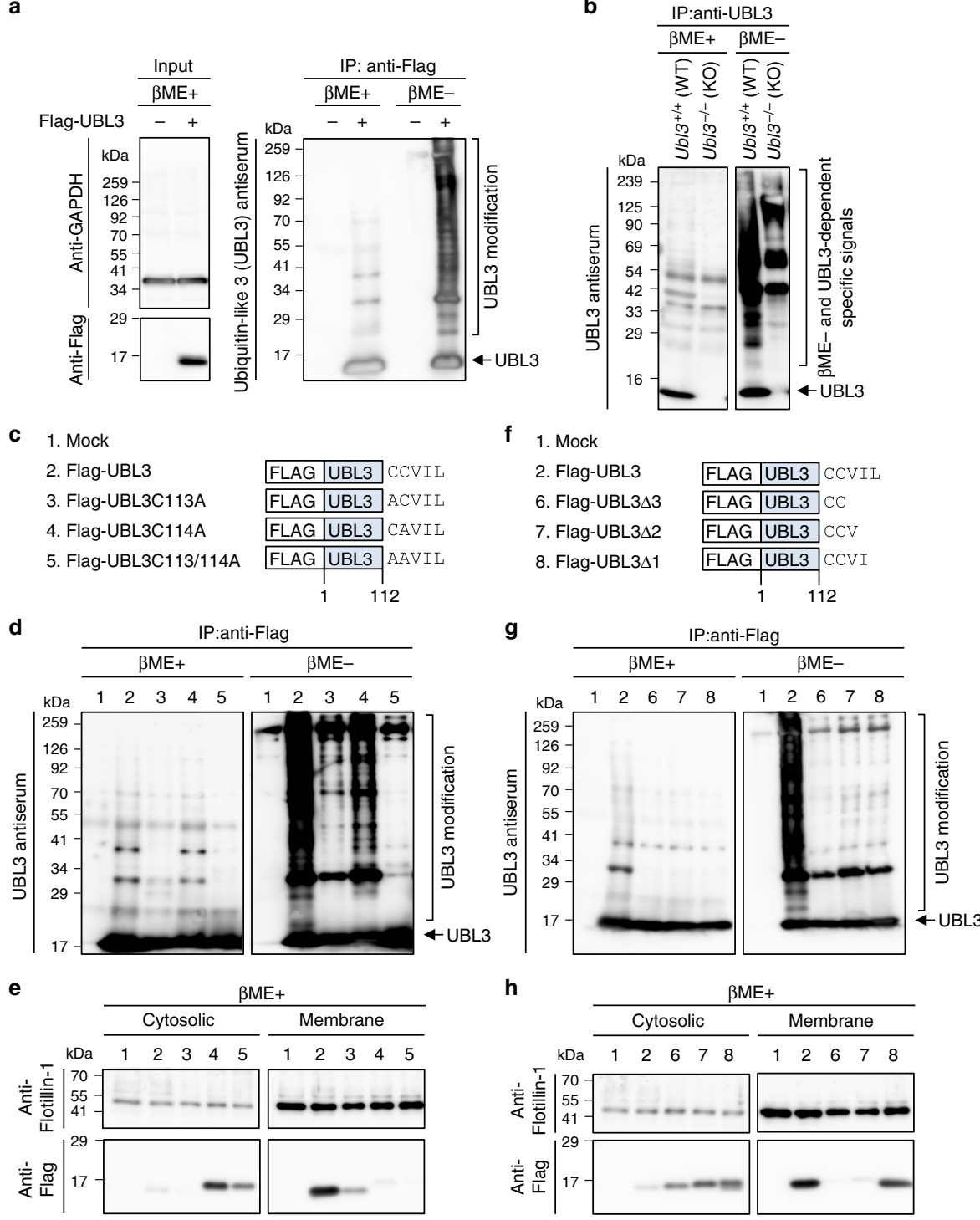

**Fig. 1** Analysis of UBL3 as a post-translational modification factor. **a** UBL3-dependent posttranslational modification was detected by immunoprecipitation (IP) with anti-Flag antibodies from MDA-MB-231 cells transfected with Flag-UBL3, followed by western blotting with UBL3 antiserum (right panel). **b** The tissue extracts from the cerebral cortex of WT and *Ubl3* KO mice were immunoprecipitated with anti-UBL3 antibodies. **c**, **f** Schematic structures of Flag-tagged wild-type or mutant UBL3. **d**, **g** Detection of UBL3 modification in these cells. IP products were boiled without 2-mercaptoethanol (βME-). A portion of the samples was treated with 2-mercaptoethanol (βME+). **e**, **h** Subcellular localisation of UBL3 in MDA-MB-231 cells transfected with Flag-UBL3 (wild-type and mutants). Twenty μg per lane

UBL3 was more concentrated in the 100K fraction than in the 2K and 10K fractions (Fig. 3a). We verified that sEVs purified from the 100K pellet contained nanometre-sized vesicles ranging from 50 to 100 nm in diameter by negative EM staining (Supplementary Fig. 5a). We found that the majority of UBL3 was present in sEVs containing CD63 (Supplementary Fig. 5b). Additionally, we found that UBL3 was packaged within the sEVs (Supplementary Fig. 5c). In order to determine whether endogenous UBL3 was released in sEVs, we prepared primary cultured cells from wild-type and *Ubl3* KO mice, purified sEVs from the

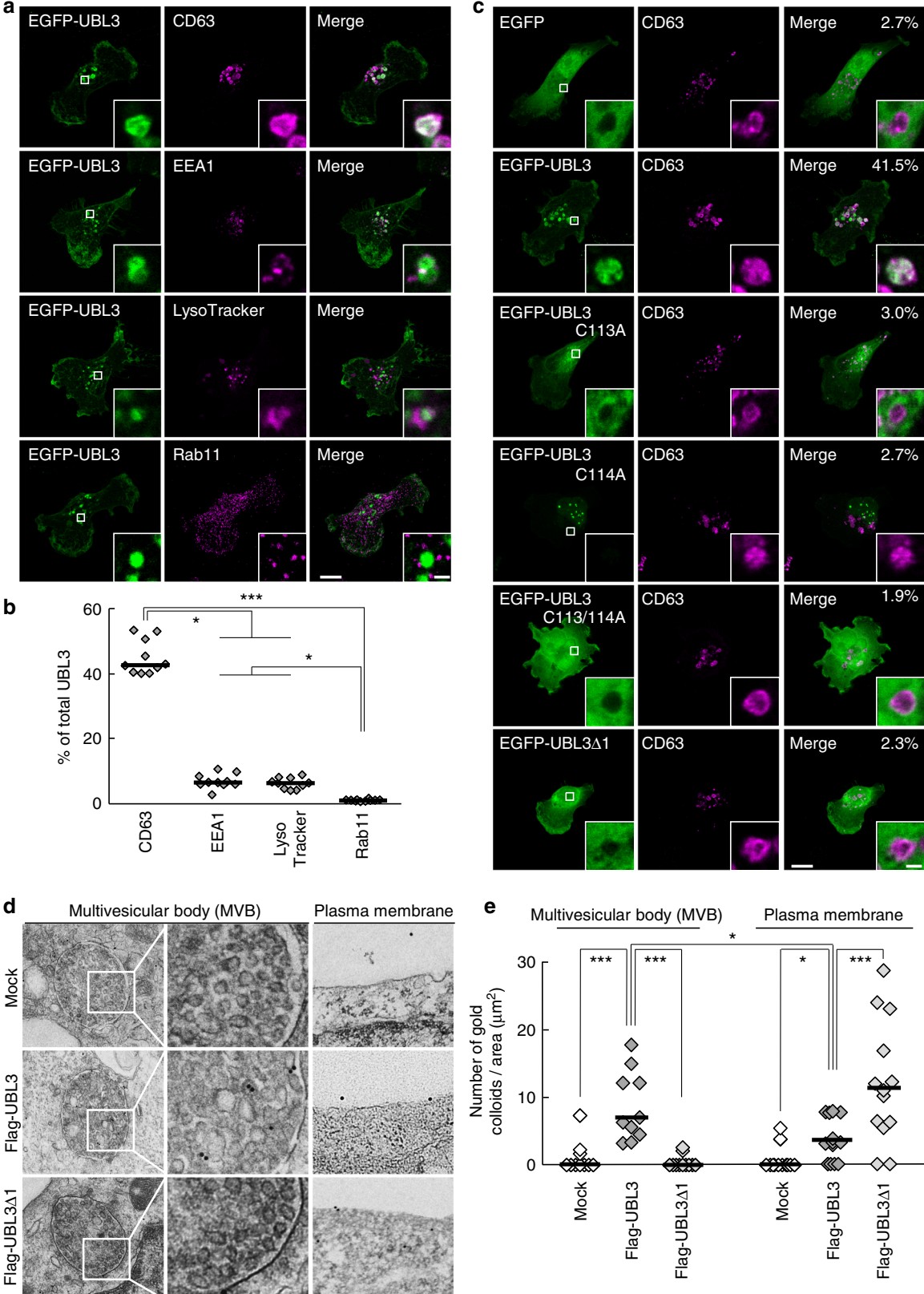

culture supernatant, and blotted them using anti-UBL3 antibodies. Endogenous UBL3 was only found in sEVs from wild-type mice (Supplementary Fig. 5d). Finally, UBL3 modification was observed not only in cell lysates but also in sEVs (Supplementary Fig. 5e). In order to clarify that UBL3 modification is indispensable for sorting to sEVs, the presence of UBL3 and its mutants in 100K sEV was studied. Among these, UBL3 cysteine mutants were not sorted to sEV fractions. On the other hand, UBL3Δ1 was detected at much reduced levels compared with those of the wild-type UBL3 (Fig. 3b). Purified 100K pellets

**Fig. 2** The localisation of UBL3 to MVBs depends on UBL3 modification. **a** Representative projected images of MDA-MB-231 cells transfected with EGFP-UBL3 and co-stained with markers for MVB (CD63, n = 10), early endosome (EEA1, n = 10), lysosome (LysoTracker, n = 10), or recycling endosome (Rab11, n = 10). The regions in the dotted box are shown as a single confocal image in the inset. Scale bars, 10 and 1 µm. **b** Quantitative analysis of EGFP-UBL3 fluorescence intensity in a. *, p < 0.05; ***, p < 0.001 by Kruskal–Wallis/Dunn´s multiple-comparisons test. **c** Representative images of MDA-MB-231 cells transfected with EGFP-UBL3 (wild-type, n = 5; mutants, n = 5) and co-stained with CD63 values shown as % of total UBL3. Scale bars, 10 and 1 µm. **d** Immune-EM images of wild-type UBL3 and UBL3Δ1 in MDA-MB-231 cells. Scale bars, 500 nm (left and right panels) and 200 nm (middle panel). **e** Quantification of the numbers of gold colloids per area in d. MVB, n = 10; Plasma membrane, n = 10. *, p < 0.05; ***, p < 0.001 by two-tailed Student's and Welch's t-tests

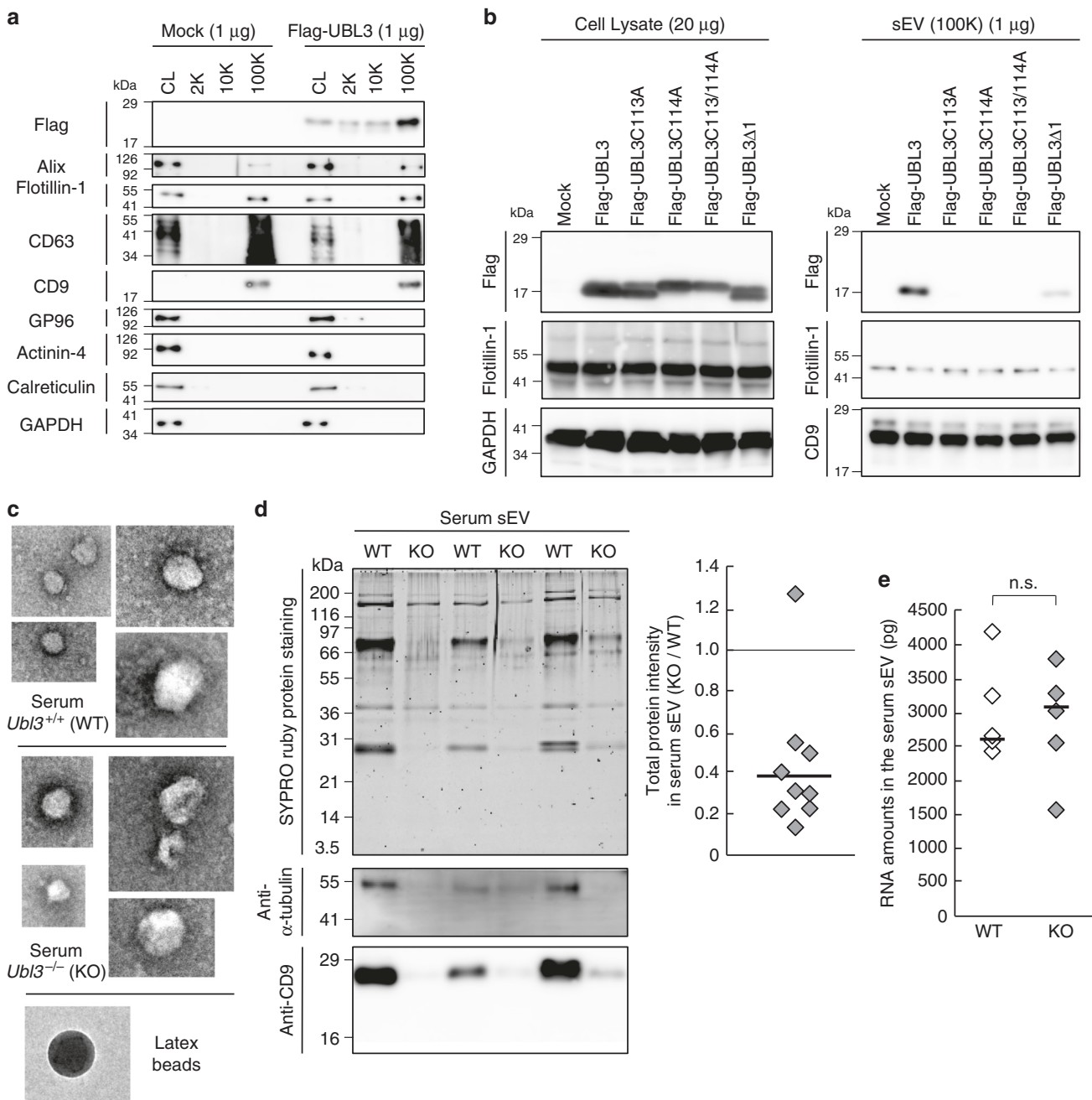

**Fig. 3** The level of total proteins in the sEVs is reduced in *Ubl3* knockout mice. **a** The cell lysate (CL) and pellets from the conditioned medium of MDA-MB-231 cells transfected with 3xFlag-UBL3 vectors were blotted with various antibodies. **b** The presence of UBL3 and its mutants in sEVs. **c** Electron microscopic analyses of purified sEVs by negative staining. Scale bars, 100 nm. **d** Upper left panel, protein staining for the sEVs from the serum in WT and *Ubl3* KO mice. Lower panels, purified serum sEVs were blotted with anti-tubulin and CD9 antibodies with βME. Right panel, relative intensity of total proteins in serum sEVs. n = 9 pairs. **e** Total RNA levels in the serum sEVs. WT, n = 5; KO, n = 5. n.s. p > 0.05 by Mann–Whitney test. a,b, βME + condition: Flag, Flotillin-1, GP96, Actinin-4, Calreticulin, GAPDH, and Alix antibodies. βME- condition: CD63, and CD9 antibodies

contained both exosomes and non-exosomal sEVs. The latter include vesicles directly secreted from the plasma membrane[21,22]; this explains why UBL3Δ1, which shows plasma membrane but not MVB localisation, was detected in the 100K pellets in small amounts (Figs. 1h, 2d, e, 3b). This finding indicates that the UBL3 modification is important for sorting of UBL3 to sEVs. Furthermore, when exosome release was inhibited by *Rab27a* shRNA[23], the levels of UBL3 modification products were increased (Supplementary Fig. 5f). The above results indicate that UBL3 modification is required for localisation of UBL3 in MVBs, and that the majority of UBL3 is secreted in sEVs originating from the fusion of MVBs released as exosomes. As UBL3 was localised in the plasma membrane, UBL3 modification may also influence non-exosomal sEVs.

**The level of total proteins in the sEVs is reduced in *Ubl3* knockout mice**. To elucidate the functional role of UBL3 in the sorting of proteins to sEVs, we quantified the protein contents of purified sEVs derived from *Ubl3*-KO mouse sera. The genotype had no effect on the concentration of sEV particles or the average vesicular diameter (Fig. 3c, Supplementary Fig. 6a, b). Intriguingly, the total level of sEV proteins in the sera from the *Ubl3* KO mice was 60% lower than that from the wild-type mice, although the total protein content in the serum and plasma were not different (Fig. 3d, Supplementary Fig. 6c). We did not find any significant difference in the RNA profiles and RNA levels (Fig. 3e and Supplementary Fig. 6d, e). Specific sEV-enriched miRNAs levels were not statistically different (Supplementary Fig. 6f). These results indicate that the *Ubl3* KO mice produce sEVs with reduced proteins, but with normal levels of miRNAs in the serum. In other words, the sorting of proteins in the serum sEVs is influenced by UBL3.

**UBL3 modification influences the sorting of proteins to the sEV**. For a better understanding of the physiological function of UBL3 modification, we performed comprehensive proteomics analysis to identify proteins that interacted with UBL3 in a manner dependent on the two C-terminal cysteine residues. Lysates prepared from 3xFlag-UBL3 expressing in MDA-MB-231 cells were purified with Flag antibody-beads, subjected to on-bead digestion, and analysed by nanoflow liquid chromatography tandem mass spectrometry (LC-MS/MS) gradients on a Q Exactive HF mass spectrometer (Fig. 4a). The dataset covered 3882 proteins in total at a false discovery rate (FDR) of 1%. The average of Pearson correlation coefficients was 0.988 within triplicate pull-downs. Label-free quantification (LFQ)-based unsupervised hierarchical clustering of significantly regulated proteins (ANOVA, FDR = 0.05 & S0 = 1) revealed 1447 possible UBL3-interacting proteins that were upregulated in Flag-UBL3 compared to Flag-UBL3C113/114A or the Flag empty vector (Fig. 4a and Supplementary Table 1). Next, we grouped the UBL3-interacting proteins based on their subcellular localisation as defined by the Gene Ontology Cellular Components (GOCC). We found that 31% of proteins (454 out of 1447 proteins; *p*-value of 0.00405, Fisher exact test) were categorised as the GOCC term 'extracellular vesicular exosome' (Fig. 4b, and Supplementary Data file). In addition, the pairwise comparison of UBL3 with UBL3C113/114A revealed that 1241 proteins significantly interact with UBL3, emphasising the role of cysteine residues in UBL3 modification (two-sample Student's *t*-test, FDR = 0.05 & S0 = 1). Of these proteins, 29% (369) of them were also annotated as extracellular vesicular exosome. Moreover, Fisher exact test for the statistically significant proteins showed an enrichment for the annotation of extracellular vesicular exosome (*p*-value of 0.00828). Among these, tubulins (TUBA1A, 1B, 1C, and 4A; and TUBB, B2A, B3, B4A, B4B, B6, and B8) were

found in 3xFlag-UBL3 samples (Fig. 4c, Supplementary Fig. 7a and Supplementary Data file). Tubulins are reported to be present in sEVs and other EVs[21,24]. Endogenous tubulin alpha can be detected in sEVs with the commonly used specific antibody (DM1A); therefore we chose tubulin alpha as a model case to study sorting of endogenous proteins to sEVs via UBL3 modification. When EGFP-UBL3 introduced to MDA-MB-231 cells was immunopurified from anti-GFP-beads, tubulin was found to be modified by UBL3 and observed as shifted bands only under non-reducing conditions (Supplementary Fig. 7b). We then studied the sorting of tubulin by UBL3 in secreted vesicles prepared by differential centrifugation. Intriguingly, the tubulin signal was specifically augmented by UBL3 expression in the 100K, but not in the 2K or 10K fractions (Supplementary Fig. 7c). Under non-reducing conditions, the majority of tubulin in the 100K sEV pellet was found to have a higher molecular weight than under reducing conditions. We used UBL3 mutants to investigate the relationship between UBL3 modification of tubulin and the sorting to sEVs. The increase in tubulin in the sEVs was specific to wild-type UBL3; the UBL3C113/114A and UBL3Δ1 mutants exhibited no such effects (Supplementary Fig. 7d). We also observed that the protein levels of tubulin in the serum sEVs were reduced in *Ubl3* KO mice (Fig. 3d). We therefore conclude that UBL3 modifies endogenous tubulin posttranslationally and regulates its sorting to sEVs.

We also found that at least 22 disease-related molecules were included as UBL3-interacting proteins (Fig. 4c and Supplementary Fig. 8a). Intriguingly, some of these molecules are related to oncogenesis and tumour progression/metastasis, namely, HRAS, KRAS, TGFBR1, TGFBR2, RB1, ITGA6, ITGB4, mTOR, TSC2 and APLP2. In addition, immune-responsive molecules (IRF3 and IKBKG), mTOR signalling molecules (mTOR, RPTOR and TSC2), Notch signalling molecules (NOTCH1, NOTCH2, NOTCH3), BMP signalling molecules (BMPR1A and BMPR2), and even molecules involved in neurodegenerative/neuronal diseases (PSEN1, ATXN10, HIP1R, APLP2 and NPC1) are identified as UBL-interacting proteins. Among these, Ras family members are proto-oncogenes[25] and have been reported to be enriched in sEVs[26]. Therefore, we chose H-Ras as a model protein to study the sorting to sEVs via UBL3 modification. We found that exogenous wild-type Ras and constitutively active oncogenic RasG12V mutant[25] interacted with UBL3, and were observed as shifted bands only under non-reducing conditions (Supplementary Fig. 8b). Moreover, exogenous wild-type Ras proteins were more sorted to sEV by UBL3 (Supplementary Fig. 8c). We also found that wild-type UBL3 but not UBL3C113/114A enhanced sorting of RasG12V to sEVs (Fig. 4d).

To examine whether sEVs encapsulated with UBL3 and oncogenic RasG12V caused the activation of Ras signalling in the recipient cells, we added PKH67-labelled sEV purified from MDA-MB-231 cells that were transfected with RasG12V and either mock, 3xFlag-UBL3 or 3xFlag-UBL3C113/114A to recipient MDA-MB-231 cells. As an indicator of Ras activation in the recipient cells, we measured the phosphorylation level of ERK, which is a downstream signalling molecule[25]. Phosphorylated ERK (pERK) activity in PKH67-labelled sEV-incorporated cells was examined by antibody staining. We found that purified sEVs from MDA-MB-231 cells transfected with wild-type UBL3 and RasG12V significantly enhanced the level of pERK in the recipient cells compared to sEVs from UBL3C113/114A and RasG12V transfected cells (Fig. 4e, f). These results indicated that increased sorting of RasG12V to sEVs by UBL3 modification enhanced activation of Ras signalling in the recipient cells.

We also performed a comparative analysis using other UBLs. Unlike UBL3, other UBLs, including ubiquitin, SUMO1, and SUMO2, were not enriched either in MVBs or sEVs (Fig. 5a–c).

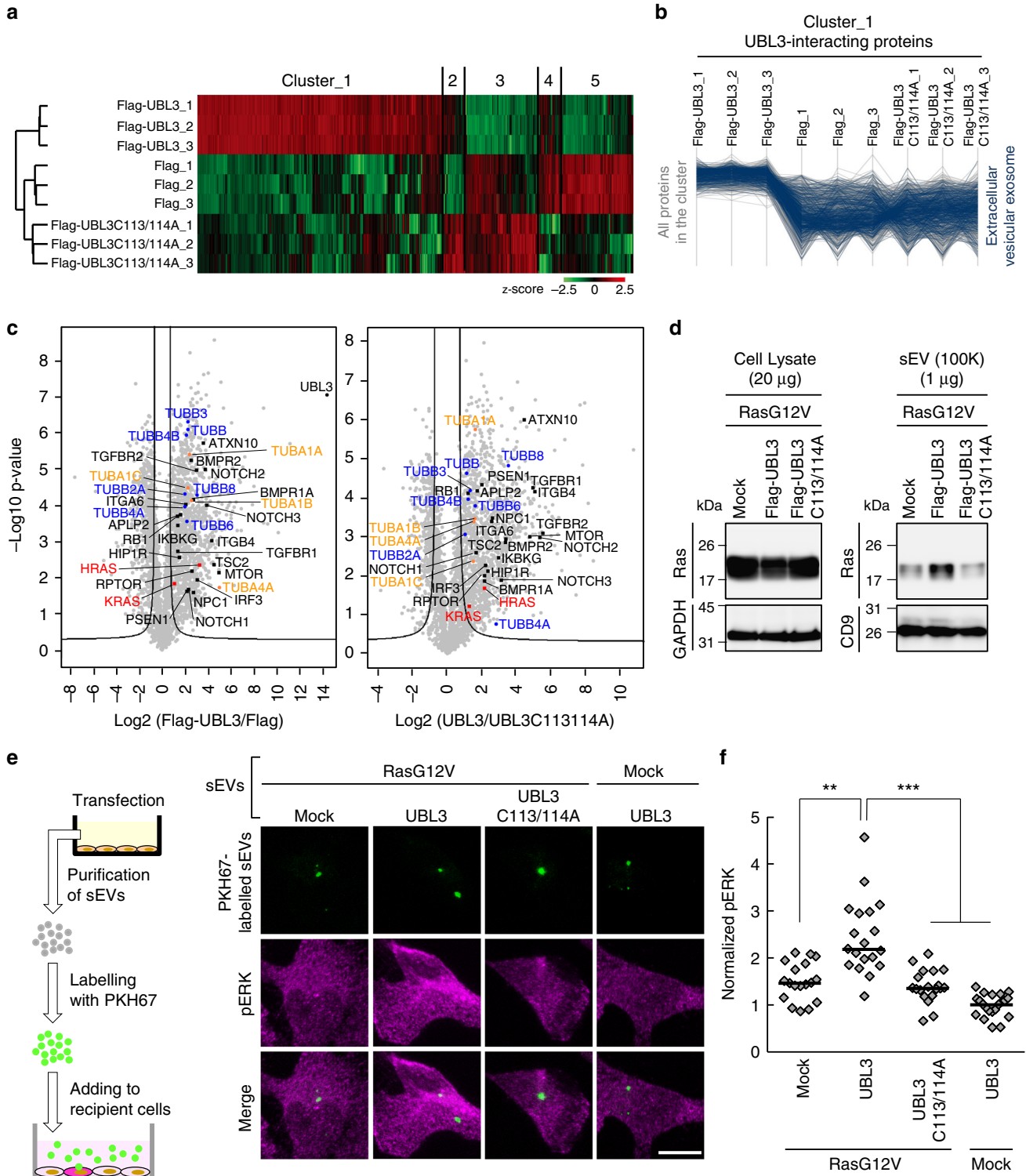

Finally, we attempted to determine whether UBL3 serves as a useful tag for protein delivery to sEVs. When EGFP or biotinylated-protein tagged with UBL3 was introduced to MDA-MB-231 cells, EGFP-UBL3 and biotinylated-UBL3 were found in sEVs (Fig. 6). These observations confirmed that UBL3 serves as a tag for the delivery of proteins to sEVs.

## Discussion

In the current study, we characterised UBL3 as a PTM factor. The glycine residues at the C-termini of ubiquitin, SUMO, and Nedd8

are covalently attached to lysine residues in target proteins in general[15,18,27]. In contrast, our data suggest that UBL3 modifies target proteins by disulfide bonding through cysteine residues at its C-terminus; therefore although UBL3 has a ubiquitin-like domain, UBL3 modification is found to be completely different from conventional ubiquitin and ubiquitin-like modifications. From the result of UBL3Δ1 mutant (Fig. 1g), the membrane localisation of UBL3 alone is not sufficient for the UBL3 modification. Thus, in addition to CAAX motif for the membrane localisation in the C-terminal of UBL3, it is possible that

**Fig. 4** UBL3 modification influences the sorting of proteins to the sEV. **a**, **b** Heat map of z-scored LFQ intensities of the significantly regulated proteins (ANOVA, FDR = 0.05 & S0 = 1) in all three conditions (3xFlag-UBL3, 3xFlag-UBL3C113/114A or 3xFlag empty vector) revealed the UBL3 interacting proteins (Cluster 1). The colour key denotes normalised protein abundances (z-score). Profiles of all proteins (1447) found in Cluster 1 are shown and 454 of those (31%) with 'extracellular vesicular exosome' annotations are highlighted in dark blue. **c** Volcano plots showing the *p*-values vs. the log2 protein abundance differences in Flag-UBL3 compared with either Flag-UBL3C113/114A or Flag empty vector. The significance cut-off is based on an FDR = 0.05 and S0 = 1. Disease-related molecules, black colour. HRAS and KRAS, red colour. TUBA1A, 1B, 1C, and 4A, orange colour. TUBB, B2A, B3, B4A, B4B, B6, and B8, blue colour. **d** sEV pellets were blotted with anti-Ras antibodies. **e** Images of phosphorylated ERK (pERK) in PKH67-labelled sEV-incorporated MDA-MB-231 cells. Purified sEVs from the conditioned medium of MDA-MB-231 cells transfected with RasG12V and either mock, 3xFlag-UBL3, or 3xFlag-UBL3C113/114A or with mock and 3xFlag-UBL3 were labelled with PKH67 dye (green) and added to MDA-MB-231 cells. Scale bars, 10 μm. **f** Each plot shows pERK fluorescence in PKH67-labelled sEV-incorporated cells normalised to the average pERK values in the two neighbouring PKH67-labelled sEV-unincorporated cells from each image in e. RasG12V-mock, n = 18; RasG12V-UBL3, n = 19; RasG12V-UBL3C113/114A, n = 19; Mock-UBL3, n = 21. **, *p* < 0.01; ***, *p* < 0.001 by Kruskal–Wallis/Dunn's multiple-comparisons test

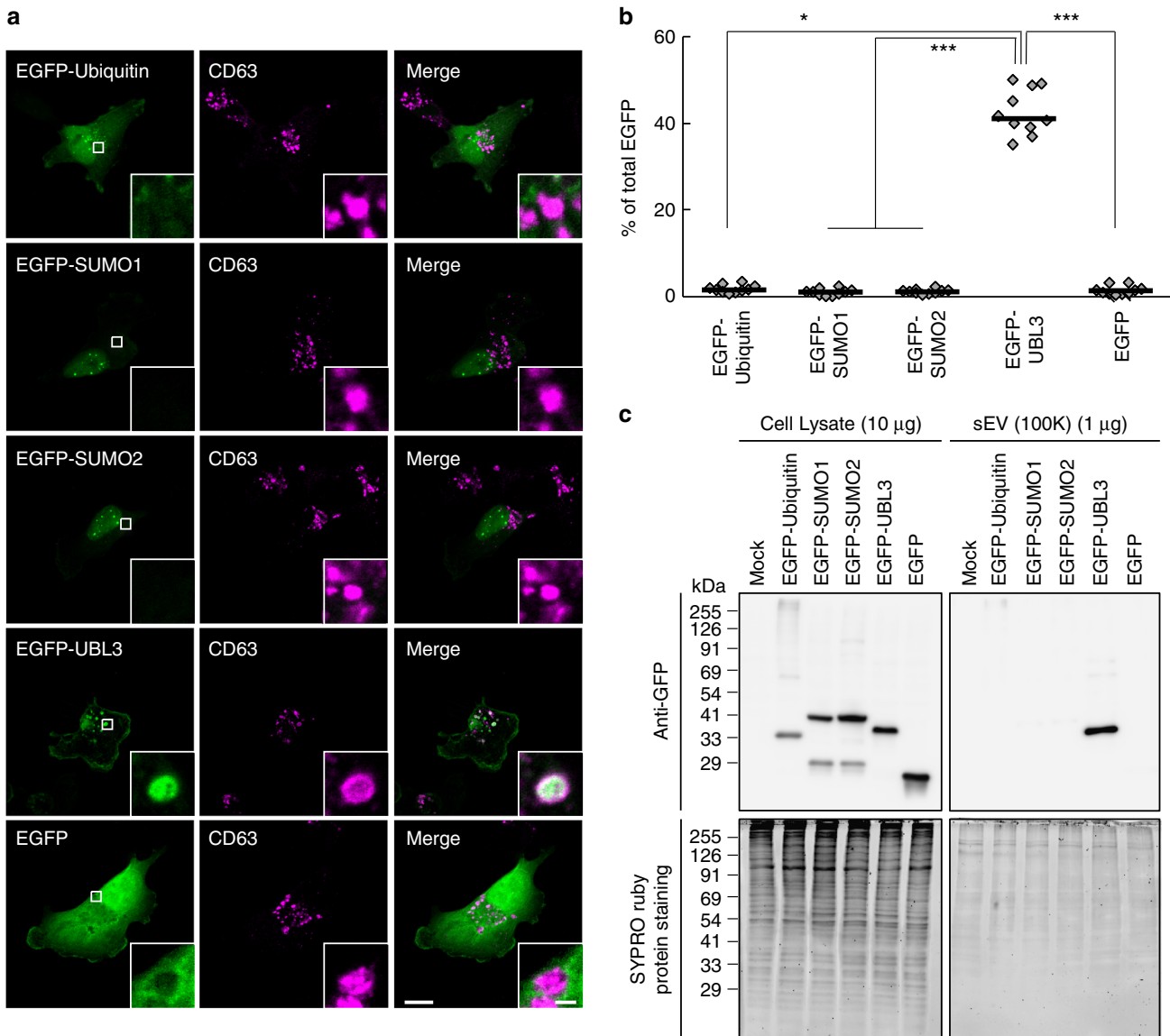

**Fig. 5** Subcellular localisation and sorting to sEV of UBLs. **a** Representative images of MDA-MB-231 cells transfected either with EGFP-ubiquitin, -SUMO1, -SUMO2 or -UBL3 and co-stained with markers for MVBs (CD63). Scale bars, 10 and 1 μm. **b** Quantitative analysis of EGFP-ubiquitin, -SUMO1, -SUMO2, and -UBL3 fluorescence intensity in a. EGFP-ubiquitin, n = 10; EGFP-SUMO1, n = 10; EGFP-SUMO2, n = 10; EGFP-UBL3, n = 10; EGFP, n = 10. *, *p* < 0.05; ***, *p* < 0.0001 by Kruskal–Wallis/Dunn's multiple-comparisons test. **c** EGFP-UBL3, but not EGFP, EGFP-ubiquitin, EGFP-SUMO1, or EGFP-SUMO2, was preferentially enriched in the sEVs of the cell culture media. Before sample loading, the samples were boiled with βME

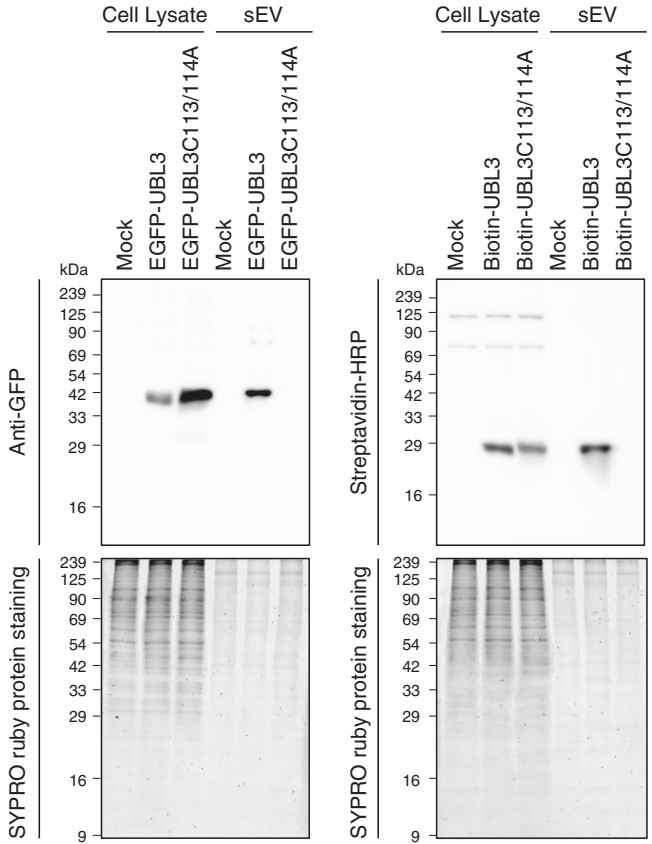

**Fig. 6** EGFP and biotinylated protein tagged by UBL3 are sorted to the sEVs. EGFP-UBL3 (left panel) and biotinylated protein tagged by UBL3 (right panel), but not EGFP-UBL3C113/114A or biotinylated protein tagged by UBL3C113/114A were preferentially accumulated in the sEVs of the cell culture media. After the western blot analysis, gels were stained with SYPRO Ruby (lower panel). Before loading, the samples were boiled with βME. The same amounts of protein were loaded on the gels (Cell Lysate, 20 μg per lane; sEVs, 1 μg per lane)

important amino acid sequences for the UBL3 modification in the C-terminal region exist. In the future UBL3 study, it is important to analyse molecules that recognise the C-terminal sequence for UBL3 modification.

Additionally, we report that UBL3 modification is essential for the sorting of UBL3 to MVBs and sEVs. We identify 1241 UBL3-interacting proteins dependently on the C-terminal two cysteine residues, 29% of which are annotated as 'extracellular vesicular exosome'. We also observed a 60% reduction in the total sEV proteins in the sera of *Ubl3* KO mice (Fig. 3d). This result suggested that UBL3 may be involved in the sorting of more than half of all exosomal proteins. It is possible that UBL3 binding to a particular protein is transient, thereby permitting identification of only a subset of UBL3-modified proteins in the comprehensive proteomics analyses of cell lysates. Alternatively, UBL3 modification may be required for only a limited number of target proteins that participate in large protein-protein interaction networks for targeting to exosomes; however, this idea must be confirmed by further investigation.

Specific proteins in sEVs play an important role in tumour progression and organotropic metastasis[5,28,29]. Neurodegenerative disease-related proteins, such as amyloid beta, tau, α-synuclein, and prions, are also packaged inside sEVs and spread in the brain[6–8,10]. In the current study, we identified 1241 UBL3-interacting proteins that were dependent on the two C-terminal

cysteine residues, including at least 22 disease-related molecules (Fig. 4c), and showed that oncogenic RasG12V proteins were more sorted to sEVs by UBL3 modification (Fig. 4d). These sEVs bearing the oncogenic proteins affected ERK phosphorylation in the recipient cells (Fig. 4e, f). Therefore, the inhibition of UBL3 modification also could be a therapeutic target for sEV-related disorders. Our results also indicate that extraneous proteins are sorted to sEVs by UBL3 tagging (Fig. 6). Thus, for any other type of sEV-related diseases, UBL3 could serve as a useful tool for the generation of sEVs that are modified to act as vehicles for therapeutic cargos.

## Methods

**Cell culture.** MDA-MB-231-luc-D3H2LN breast cancer cells (Xenogen Co, CA), HEK293T cells (RIKEN Cell Bank, Japan), and HeLa cells (RIKEN Cell Bank, Japan) were cultured in Roswell Park Memorial Institute (RPMI) 1640 medium (GIBCO, 11875-093) or Dulbecco's modified Eagle's medium (DMEM, GIBCO, 11965-092) with PS (100 units of penicillin G per mL, 10 μg of streptomycin sulphate per mL) and 10% foetal bovine serum (FBS). Cultures were incubated in a 5% $CO_2$ at 37 °C. All the cell lines were routinely tested for mycoplasma contamination (Biological Industries, EZ-PCR Mycoplasma Test kit).

Following previous papers[30,31], we prepared primary cells from the adipose tissue-derived stromal vascular fraction (SVF) and myoblasts from wild-type and *Ubl3* KO mice. Briefly, to prepare primary SVF cells, inguinal and visceral adipose tissues were excised from mice and digested with 3.33 mg mL$^{-1}$ type II collagenase (Nitta-gelatin, type L) with gentle rocking in Hank's balanced salt solution for 60 min at 37 °C. After sequential filtration through 40- and 100-μm filters (BD), the isolated SVF cells were suspended in FBS and cultured in DMEM (GIBCO, 11965-092) with 10% FBS and PS. Myoblasts were cultured using the following growth medium in a culture dish coated with collagen type I (IWAKI, 44020-10) for 3–5 days: Ham's F-10 Nutrient Mix (GIBCO, 11550-043) containing 20% FBS, 2.5 ng mL$^{-1}$ bFGF (Katayama Chemical Industries, 161-0010-5), L-glutamine, penicillin, and streptomycin (Thermo Fisher Scientific, PS-glutamine). For myotube differentiation, the medium was changed to a differentiation medium (5% horse serum in DMEM [GIBCO, 11995-065]) with PS-glutamine for 5 days.

**Plasmid construction.** Various expression vectors were subcloned either to pcDNA3 (Invitrogen), pEGFPC (Clontech), or pcDNA6/BioEase (Invitrogen) using conventional molecular biology techniques and PCR[14,31,32]. The sequence of *Rab27a* shRNA (5'-GCTGCCAATGGGACAAACATA-3') was used[23] and inserted into pcDNA 6.2-GW/miR vector (Invitrogen).

In the Ras experiments, we used H-Ras expression vector set (Clontech, pCMV-Ras vector, and pCMV-RasV12 vectors). GFP-ubiquitin was a gift from Nico Dantuma (Addgene plasmid # 11928)[33].

**Establishment of *Ubl3*$^{-/-}$ (knockout; KO) mice.** The targeting strategy is illustrated in Supplementary Fig. 2a. The targeting construct was linearised by SpeI digestion. CCE (129/Sv/Ev) (a gift from Dr. E. Robertson) embryonic stem (ES) cells were electroporated using a SpeI-linearised targeting vector and selected in G418. Heterozygous ES cells were injected into the blastocysts of C57BL/6J strain mice to generate germ-line chimeras. Chimeric mice were backcrossed with C57BL/6J mice for 10 generations. Genotypes were determined by Southern blotting or by PCR using the following three primers: *Ubl3*-KO-3', 5'-ACCCA GGTCCTCATGCATCGGTAGA-3'; *Ubl3*-KO-5', 5'-CCACCCACTGCCTTTC CCAGAAAC-3'; and *Ubl3*-KO-Neo, 5'-GCTGCAGGGTCGCTCGGTGTT-3'. All procedures related to the care and treatments of animals were in accordance with institutional and National Institutes of Health guidelines and were approved by the Animal Care and Use Committee at Fujita Health University.

**Antibodies.** To obtain the rabbit anti-UBL3 antibody, 1.5 mg of glutathione S-transferase (GST)-fused mouse UBL3 protein expressed in bacteria was injected four times each into nine rabbits. Recombinant, exogenous, and endogenous UBL3 proteins were specifically recognised in only one rabbit's serum. The serum was passed through the GST column three times to remove non-specific antibodies. We called the resulting flow-through (FT) the UBL3 antiserum. To obtain a more specific antibody for the UBL3 protein, a polyclonal UBL3 antiserum was passed through the GST column. The resultant FT was further passed through a column conjugated with a brain lysate from *UBL3* KO mice to exclude the recognition of other UBLs. The FT was further purified using a GST-UBL3 column. We called this eluate the anti-UBL3 antibody. The antibodies used in the experiments were anti-GAPDH (Cell Signalling, 14C10, 1:1000), Flag-M2 (Sigma, F3165, 1:1000), Flotillin-1 (BD, Clone 18, 1:250), CD63 (Invitrogen, Ts63, 1:1000 for WB, 1:10000 for IF), EEA1 (BD, 610456, 1:5000), Rab11 (BD, 610656, 1:150), COXIV (Cell Signalling, 4850, 1:1000), Calnexin (Enzo, ADI-SPA-860, 1:500), GM130 (BD, 610822, 1:100), PMP70 (Thermo, PA1-650, 1:50000), Lamin B1 (Abcam, ab16048, 1:800), GP96 (Enzo, 9G10, 1:500), alpha Actinin-4 (GeneTex, C2C3, 1:250),

Calreticulin (Cell Signalling, 2891, 1:500), Alix (Abnova, H00010015-B02, 1:300), CD9 (Affymetrix, 14-0091, 1:500; Invitrogen, Ts9, 1:500), alpha-Tubulin (Sigma, DM1A, 1:1000), GFP (MBL, 598, 1:1000; Thermo, A11122, 1:1000), pan-Ras (Cell Signalling, 3339, 1:1000; Millipore, MABS195, 1:500), Phospho-ERK (Cell Signalling, 4370, 1:200), horse radish peroxidase (HRP)-conjugated Streptavidin (Invitrogen, 19534-050, 1:5000), Alexa 488- or 555-conjugated antibodies (Thermo), and HRP-conjugated secondary antibody (eBioscience, Cell Signalling, and Abcam). In a differential centrifugation experiment, we used the antibodies against the following markers: sEV-markers, CD63 and CD9; multiple EV marker, Flotillin-1 and Alix; and non-sEV-markers, GP96, Actinin-4, Calreticulin, and GAPDH. Chemiluminescence detection was performed with an image analyser LAS-4000 with Image Reader LAS-4000 (Ver. 2.0) and MultiGauge software (FUJIFILM). Uncropped version of each image is shown in Supplementary Fig. 9.

**Immunoprecipitation.** For the detection of UBL3 modification, MDA-MB-231, HEK293T, and HeLa cells were transfected with DNA using Lipofectamine 2000 (Invitrogen). After 24 or 72 h (Supplementary Fig. 5f), cells were washed and collected by scraping into ice-cold PBS, pelleted by centrifugation at $400 \times g$ for 1 min at 4 °C, and lysed with 1% Triton buffer (50 mM Tris-HCl [pH 7.4], 100 mM NaCl, and 1% (v/v) Triton X-100) for 20 min at 4 °C. Crude nuclei and unbroken cells were excluded by centrifugation at $20,000 \times g$ for 5 min. In the experiment using hypotonic buffer, after being washed in ice-cold PBS, cells were resuspended in hypotonic buffer (10 mM HEPES [pH 7.2], 10 mM KCl, 1.5 mM MgCl$_2$, 0.1 mM EGTA) and disrupted by a Dounce homogeniser. Crude nuclei and unbroken cells were excluded by centrifugation at $800 \times g$ for 5 min at 4 °C. For pre-clearing, the cell lysates were incubated with 20 µL of Protein G-sepharose beads (GE Healthcare) for 1 h at 4 °C with rotation. The resulting supernatant was incubated with 2 µg of anti-Flag antibody in 15 µL of Protein G-sepharose beads for 10 h at 4 °C with rotation, or 20 µL of GFP-Trap (ChromoTek) for 1 h at 4 °C without pre-clearing. After the beads were washed four times with ice-cold wash buffer (50 mM Tris-HCl [pH 7.4], 100 mM NaCl), 50 µl of 2× sample buffer (100 mM Tris–HCl [pH 6.8], 4% SDS, 20% glycerol, and 0.01% bromophenol blue, without 2-mercaptoethanol) was added to the beads, and the beads were boiled for 3 min (βME −). A portion of the samples was treated with 2-mercaptoethanol and boiled for 3 min (βME+). These samples were blotted with UBL3 antiserum (1:100 dilution). For the detection of endogenous UBL3 modification, 50 mg of frozen tissue was pulverised (BMS, ShakeMan2) and homogenised with RIPA buffer (50 mM Tris-HCl [pH 8.0], 150 mM NaCl, 1% Nonidet P-40, 0.5% sodium deoxycholate, 0.1% SDS). For pre-clearing, the cell lysates were incubated with 40 µL of Protein G-sepharose beads for 1 h at 4 °C with rotation. The resulting supernatants were incubated with anti-UBL3 antibody (1:1000 dilution) in 20 µL of Protein G-sepharose beads for 10 h at 4 °C with rotation.

The method for the immuno-isolation of sEVs using anti-CD9 or CD63 antibodies was referred to from Kowal J et al.[21]. Briefly, before performing the immuno-isolation, anti-CD9 (Millipore, MM2/57, 1:10), anti-CD63 (BD, H5C6, 1:50) and control mouse polyclonal IgG (Millipore, 12-371, 1:100) incubated with 100 µL of protein A magnetic beads (Thermo, 88845) for 16 h at 4 °C with rotation. The resulting protein A magnetic beads were washed three time with ice-cold 0.001% Tween/PBS, resuspended in 500 µL of ice-cold 0.001% Tween/PBS. Two µg purified sEVs were added to the magnetic beads and incubated for 16 h at 4 °C with rotation. Bead-bound sEVs were collected and washed three time in 500 µL of ice-cold 0.001% Tween/PBS .

For the detection of UBL3 inside the sEVs, sEVs were purified from the media of biotinylated-tagged UBL3-transfected MDA-MB-231 cells, washed with PBS or SDS buffer (50 mM Tris-HCl [pH8.0], 150 mM NaCl, 1% sodium deoxycholate, 1% NP-40, 2% SDS), and then treated with streptavidin beads (Thermo, 29202). The pulldown (PD) and flow-through (FT) fractions were analysed by western blotting with streptavidin-HRP or anti-CD63 antibodies.

**Miscellaneous procedures.** Tissue homogenisation and western blotting analysis were performed using conventional methods[14,31,32]. Isolation of membrane and cytoplasmic protein fractions were performed following the manufacturer's protocols (Fermentas, ProteoJET; Thermo, Mem-PER Plus kit). RNA purification, reverse transcription reaction, and real-time qPCR analysis (Takara Bio, Thermal Cycle Dice (TP800), Thermal Cycle Dice Real Time System Ver. 5. 11B) were performed by conventional methods[14,31,32]. Briefly, to detect miRNAs, cDNA was generated using an miScriptReverse Transcription Kit (QIAGEN). Expression of miR-16, miR-23 and miR-21 was detected using an miScript SYBR Green PCR Kit (QIAGEN) with the following primers: 5'- TAGCAGCACGTAAATATTGG-3' for miR-16, 5'-ATCACATTGCCAGGGATTTCC-3' for miR-23 and 5'-TAGCTTAT CAGACTGATGTTGA-3' for miR-21.

**Immunocytochemistry.** MDA-MB-231 cells were co-transfected with GFP-UBL3 (wild-type or mutant) and iRFP670 (morphological marker) using Lipofectamine 2000. After 24 h, cells were fixed with 4% PFA/PBS for 15 min; quenched with 0.1 M glycine/PBS; permeabilized/blocked with 0.2% BSA, 2% goat serum, and 0.05% saponin/PBS for 1 h; incubated with primary antibodies for 18 h at 4 °C and secondary antibodies for 1 h; and mounted with Aqua-Poly/Mount (Polysciences). Lysosomes were labelled with 50 µM LysoTracker Red (Thermo) for 1 h at 37 °C.

Confocal images were acquired with a 63× objective lens (NA 1.4) on a confocal laser microscope (Carl Zeiss, LSM780-DUO-NLO, ZEN 2012 SP1 Ver. 8. 1. 0. 0). Representative images projected from several 0.5 µm-interval sections were shown, each with a single confocal image in the inset. Fluorescence intensity and cell morphology were measured with Fiji (Image J). The percentage of UBL3 was calculated as: (%) = (the summation of the GFP-UBL3 signal in each organelle area in each confocal section)/(the summation of the GFP-UBL3 signal in the total cell area in each confocal section) × 100.

**Transfer of PKH67-labelled sEVs.** Purified sEVs from MDA-MB-231 cells were labelled using a PKH67 Green Fluorescent kit (Sigma, PKH67GL-1KT). sEVs were incubated with 2 µM PKH67 for 5 min, and then washed five times using a 100-kDa filter (Sartorius, VN01H42) to remove excess dye. MDA-MB-231 cells had been seeded at a density of $1.2 \times 10^5$ cells per well onto poly-L-lysine coated cover glasses in a 24-well plate, allowed to attach overnight, and then were incubated with PKH67-labelled sEVs (1 µgml$^{-1}$). After 12 h, MDA-MB-231 cells were fixed with 4% PFA/PBS for 15 min; permeabilized/blocked with 5% goat serum and 0.3% Triton X-100/PBS for 20 min; incubated with anti-pERK antibodies for 18 h at 4 °C and secondary antibodies for 1 h; and mounted with Aqua-Poly/Mount (Polysciences). Confocal images were acquired with a 63× objective lens (NA 1.4) on a confocal laser microscope (Carl Zeiss, LSM-710, ZEN 2009 Ver. 5. 5. 0. 0). Data were analysed by using Fiji (Image J); the values of pERK signal in PKH67-labelled sEV-incorporated cells were presented as the mean signal intensity measured over an area of 2.5–5 µm$^2$ in a region of the cytoplasm adjacent to the nucleus, and then normalised to the average values of pERK in the two neighbouring PKH67-labelled sEV-unincorporated cells, in each image.

**Immunoelectron microscopy.** 3xFlag-UBL3 (wild-type or UBL3Δ1) or mock-transfected MDA-MB-231 cells were fixed with 4% PFA/PBS for 30 min at 4 °C, and subsequently fixed with 4% PFA and 0.1% glutaraldehyde (GA)/PBS for 30 min at 4 °C. After being washed with PBS, the cells were fixed with 1% osmium tetroxide/PBS for 30 min, dehydrated through a graded ethanol series, transferred to n-butyl glycidyl ether (QY-1), and embedded in epoxy resin (TAAB, EPON812). Ultrathin sections were cut on an ultramicrotome (Reichert-Nissei, Ultracut N) at 100-nm thickness, placed on a nickel grid (VECO, 2552), etched with 5% sodium metaperiodate for 10 min, and washed with PBS. The sections were incubated with anti-Flag antibodies (1:100) for 2 h. After washing with PBS, the sections were incubated with 15 nm gold-conjugated goat anti-mouse IgG antibodies (Cyto-diagnostics, 1:50) for 1 h. After being washed with PBS, the sections were fixed in 2.5% GA/PBS for 10 min and washed with Milli-Q water to stabilise the antigen-antibody binding. The immunostained sections were further stained with uranyl acetate for 3 min and lead citrate for 1 min, and then observed under a transmission electron microscope (Hitachi, H-7650, H-7650 control Ver. 01. 10, EMIP Ver. 05. 04). The number of gold particles per unit area in the MVBs, plasma membrane, mitochondrion, and nuclear membrane were counted.

**Electron microscope analysis for sEVs.** Ultramicroscopic analysis was performed following the conventional protocol with slight modification[34]. Briefly, the sEVs isolated by the differential ultracentrifugation protocol from the conditioned medium of MDA-MB-231 cells were mixed with 4% PFA/PBS in equal amounts, then deposited on electron microscopy (EM) grids (Nisshin EM, Excel Support Film). The sEVs on the EM grids were washed with PBS, further fixed for 5 min with 1% GA/0.1M phosphate buffer, and washed eight times every 2 min with ultrapure water. The EM grid was negatively stained with 10% EM stain (Nisshin EM) and analysed using a transmission electron microscope.

**Isolation of EVs and RNA analyses.** sEVs were isolated from a conditioned medium according to the conventional protocol with slight modification[34]. Briefly, exosome-depleted FBS (centrifuged for 16 h at $100,000 \times g$) was used to prepare the conditioned medium. After incubation for 24 h, the culture medium (approximately 90 mL from $3 \times 10^7$ cells) was collected and centrifuged at $300 \times g$ for 10 min at 4 °C. The supernatant was centrifuged at $2000 \times g$ for 20 min at 4 °C. The supernatant was centrifuged again at $10,000 \times g$ for 30 min at 4 °C. To remove the cellular debris, the supernatant was filtered through a 0.22-µm filter (Millipore, Millex-GV). The sEV pellet was then harvested by ultracentrifugation at $100,000 \times g$ (Beckman, SW32Ti rotor) for 70 min at 4 °C. The sEV pellet was washed with 1 mL of PBS and collected by ultracentrifugation at $100,000 \times g$ (Beckman, TLA-110 rotor) for 60 min at 4 °C. The sEV pellet was again washed with 1 mL of PBS and collected by ultracentrifugation at $100,000 \times g$ for 60 min at 4 °C, then resuspended in PBS. For the purification of the 2K, 10K and 100K pellets, a differential ultracentrifugation protocol was used[21]. Briefly, exosome-depleted FBS was used to prepare the conditioned medium. After incubation for 24 h, the culture medium (approximately 90 mL from $3 \times 10^7$ cells) was collected and centrifuged at $300 \times g$ for 10 min at 4 °C. The supernatant was centrifuged at $2000 \times g$ for 20 min at 4 °C. The resulting pellet was called 2K pellet. The supernatant was centrifuged again at $10,000 \times g$ for 40 min at 4 °C. The resulting pellet was called 10K pellet. The supernatant was centrifuged by ultracentrifugation at $100,000 \times g$ (Beckman, SW32Ti rotor) for 90 min at 4 °C. The resulting pellet was called 100K pellet. The 100K pellet was washed with 1 mL of PBS and collected by ultracentrifugation at

100,000 × *g* (Beckman, TLA-110 rotor) for 60 min at 4 °C. The 100K pellet was again washed with 1 mL of PBS and collected by ultracentrifugation at 100,000 × *g* for 60 min at 4 °C, then resuspended in PBS. 2K and 10K pellets were washed with 30 mL of PBS, centrifuged at the same speed, and then resuspended in PBS. Filtration through 0.22-μm filters (Millipore, Millex-GV) was additionally performed before the purification of the 100K pellet. For the mouse serum, after collection, whole blood was allowed to clot by leaving it undisturbed for 30 min at RT. To remove the clot, the whole blood was centrifuged twice at 8000 × *g* for 10 min. For the preparation of mouse plasma, after collection, the whole blood was added to a heparin solution (Mochida Pharmaceutical, heparin Na 5000 units per 5 mL), rotated slowly, centrifuged at 9100 × *g* for 5 min, and the supernatant was used as the plasma.

In each experiment, the volume of serum from the *Ubl3* KO mice was matched to that of its wild-type littermates (approximately 350–500 μL). The harvested serum was immediately diluted with an equal volume of PBS and centrifuged at 2000 × *g* for 30 min at 4 °C. The supernatant was centrifuged at 12,000 × *g* for 45 min at 4 °C. The supernatant was diluted fivefold with PBS and then filtered through a 0.22-μm filter. The EV pellet was then harvested by ultracentrifugation at 110,000 × *g* (Beckman, SW55Ti rotor) for 70 min at 4 °C. The sEV pellet was washed with 1 mL of PBS and collected by ultracentrifugation at 100,000 × *g* (Beckman, TLA-110 rotor) for 60 min at 4 °C. The sEV pellet was washed again with 1 mL of PBS, collected by ultracentrifugation at 100,000 × *g* (Beckman, TLA-110 rotor) for 60 min at 4 °C, and resuspended in PBS. To quantify the total RNA amounts of serum sEVs, RNA was purified from the whole serum sEVs from each mouse using the miRNeasy Kit (QIAGEN). The purified RNA solution was measured by a Bioanalyzer 2100 system (Agilent, 2100 Expert Ver. B. 02. 08. SI648, RNA 6000 Pico kit). For real-time qPCR analysis, the purified RNA solution (40%) was reverse-transcribed into complementary DNA (cDNA), and 10% of the resulting cDNA solution for each miRNA was used in real-time qPCR analysis.

Protein concentrations of the serum, plasma, and cell lysates were measured by BCA (Thermo). To quantify the protein concentration of the 2K, 10K, and 100K pellets, each pellet was diluted with 2% SDS and measured by Micro-BCA (Thermo).

**Measuring total protein amount in the serum sEVs**. Ten percent of the purified serum sEVs from each experiment was boiled with βME containing an SDS sample buffer and separated by SDS-polyacrylamide gel electrophoresis. The resulting gel was fixed in 50% methanol and 10% acetic acid for 30 min and washed in 10% methanol and 7% acetic acid for 30 min. The gel was stained in SYPRO Ruby Gel Stain (Thermo) for 3 h in the dark. The gel was washed in 10% methanol and 8% acetic acid for 30 min, and then washed in ultrapure water for 10 min. The gel was then scanned using a fluorescence scanner (GE Healthcare, Typhoon 9400, Typhoon Scanner Control Ver. 5.0) set at 532 nm. The total protein intensity was analysed using the ImageJ software.

**Proteomics analyses**. MDA-MB-231 cells were transfected with plasmids using Lipofectamine 2000. After 24 h, cells were washed and collected by scraping into ice-cold PBS, pelleted by centrifugation at 400 × *g* for 1 min at 4 °C, and lysed with 1% Triton buffer for 20 min at 4 °C. Crude nuclei and unbroken cells were excluded by centrifugation at 20,000 × *g* for 5 min at 4 °C. For pre-clearing, the supernatants were incubated twice with 40 μL Protein G-sepharose beads (GE Healthcare, 50% slurry in lysis buffer) for 1 h at 4 °C. The supernatants were then incubated with 50 μl anti-Flag M2 affinity gel (Sigma, 50% slurry in lysis buffer) for 16 h at 4 °C. The beads were washed with 1% Triton buffer, washed three times with wash buffer (50 mM Tris-HCl [pH7.5]) to remove detergent, resuspended in 200 μL digestion buffer (2 M urea, 50 mM Tris-HCl [pH 7.4], 1 mM DTT, 5 mM iodoacetamide, 1 μg of trypsin [Promega, V5280]), and incubated by shaking at 1200 rpm for 16 h at 37 °C. The samples were acidified with 22 μL of 10% trifluoroacetic acid (TFA). The peptides contained in the supernatant were then taken by centrifugation at 4400 × *g* for 30 sec. The peptides were desalted by C18Tip (Thermo, StageTips), dried by speedvac, and then resuspended in mass spectrometry buffer (2% acetonitrile, 0.2% TFA). Peptides were separated on a 50 cm reversed-phase column (75 μm inner diameter, packed in-house with ReproSil-Pur C18-AQ 1.9 μm resin [Dr. Maisch GmbH]) with a binary buffer system of buffer A (0.1% formic acid (FA)) and buffer B (80% acetonitrile plus 0.1% FA) over a 100-min gradient (5–30% and 30–65% of buffer B for 95 and 5 min, respectively) using the EASY-nLC 1000 system (Thermo) with a flow rate of 300 nL per min. Column temperature was maintained at 50 °C. The nLC system was coupled to a Q Exactive HF mass spectrometer (Thermofisher Scientific), acquiring full scans (300–1650 *m/z*, maximum injection time 20 ms, resolution 60,000 at 200 *m/z*) at a target of 3e6 ions. The 15 most intense ions were isolated and fragmented with higher-energy collisional dissociation (HCD) (target 1e5 ions, maximum injection time 25 ms, isolation window 1.4 *m/z*, NCE 27%, under fill ratio 0.1%) and detected in the Orbitrap (resolution 15,000 at 200 *m/z*).

Raw MS files were processed within the MaxQuant environment (Ver. 1.5.9.3) using the MaxLFQ algorithm for label-free quantification and the integrated Andromeda search engine with FDR < 0.01 at the protein and peptide levels[35,36]. The search included variable modifications for oxidised methionine (M), acetylation (protein N-term), and fixed modifications for carbamidomethyl (C). Up to two missed cleavages were allowed for protease digestion. Peptides with at least

seven amino acids were considered for identification, and 'match between runs' was enabled with a matching time window of 0.7 min to allow the quantification of MS1 features which were not identified in each single measurement. Peptides and proteins were identified using a UniProt FASTA database from Homo sapiens (2015) containing 21,051 entries.

The freely available software PERSEUS (Ver. 1.5.4.1) was used to perform all statistical and bioinformatics analyses [http://www.perseus-framework.org]. The proteins identified only by site-modification or found in the decoy reverse database and the contaminants were filtered out before data analysis. MaxLFQ intensities were taken for quantification and transformed into log2 scale. Three biological replicates of each pulldown were grouped, and a minimum of three valid values was required in at least one group. Missing values were imputed based on a normal distribution (width = 0.3, down-shift = 1.8). MaxLFQ intensities were first z-scored, and samples were clustered into five clusters according to the Euclidean metric as a distance measure for column and row clustering (distance threshold = 2.8). Annotations were added from Gene Ontology (GO) for enrichment analysis with a Benjamini-Hochberg FDR threshold of 0.02. Significance was assessed using ANOVA analysis including a permutation-based FDR of 5% and an S0 value of 1. The direction of significance was also assessed by the post hoc test with 5% FDR. For pairwise comparison of groups, two-sample Student's *t*-test (two-sided) was used including a permutation-based FDR of 5% and an S0 value of 1. Fisher exact test was applied by using *p*-value for truncation with threshold value 0.02.

**Measurement of sEVs by nanoparticle tracking analysis**. Nanoparticle tracking analysis (NTA) was carried out using NanoSight LM10HS with a blue laser system (NanoSight, Amesbury) on isolated serum sEV particles diluted 25-fold with PBS[37]. The system focuses a laser beam through a suspension of the particles of interest. They were visualised by light scattering using a conventional optical microscope aligned perpendicularly to the beam axis, which collects light scattered from every particle in the field of view. A 60-s video records all events for further analysis by NTA software. The Brownian motion of each particle was tracked between frames to calculate its size using the Stokes–Einstein equation. Each sample was measured three times, and we used an averaged value for the statistical analysis.

**Statistical analysis**. Statistical analyses were carried out using Prism 6.03 (GraphPad Software), Microsoft Excel, and PERSEUS (Ver. 1.5.4.1). The Mann-Whitney test or Wilcoxon signed-rank test and Kruskal–Wallis/Dunn's multiple-comparisons test were, respectively, applied for comparisons between 2 groups and 3 or more groups except proteomics analyses. We used more than two independent samples for each experiment. Horizontal bars indicate median values for each group.

## Data availability

Proteomics raw datasets are deposited in jPOST (under accession codes JPST000315, PXD007617).

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

## Acknowledgements
The authors thank Showbu Sato, T. Hino, and R. Migishima for the generation of the *Ubl3* KO mice; Y. Oda and M. Ambai for the phylogenetic tree analysis; M. Kono, G. Niimi, N. Yamamoto, and K. Taniguchi for suggestions and technical support in electron microscope analysis; K. Otsu, Saori Sato, Y. Kato, and N. Kawamura for technical assistance; S. Kamijo, M. Matsuo, and S. Nagao for maintenance of the *UBL3* KO mice; A. Yamaguchi for genotyping the *Ubl3* KO mice; and S. Ono and A. Kato for suggestions and discussion. This work was initiated while H.A. was a research scientist at MITILS, and sEV analyses were mainly performed at FHU. This work was partly supported by JSPS KAKENHI (22700346, 26640084, JP16H06280, 17H05945, 18K07209, and 16K08599); an Intramural Research Grant (26-8 and 29-4) for Neurological and Psychiatric Disorders of NCNP; the Ministry of Education, Culture, Sports, Science and Technology (Grant No.15H05898B1); CREST from Japan Agency for Medical Research and Development; AMED (Grant No.921910520); Innovative Areas-Resource and technical support platforms for promoting research 'Advanced Bioimaging Support'; 'Fluorescence Live imaging' of the Ministry of Education, Culture, Sports, Science and Technology; and the Program for Strategic Research Foundation at Private Universities.

## Author Contributions
H.A., T.O., M.M., M.S., and K.T. contributed to the study design. H.A., N.A.-I, K. Hitachi, O.K., T.O., H.Y., T.K., K. Hatanaka., K.I., Y.Y., N.K., M.N., A.U., T.I., Y.T., H.S, and M.K. participated in data collection and interpretation. H.A., K. Hitachi, T. K., H.S., and K. Hatanaka cloned constructs. K.N. contributed new reagents/analytic tools. K. Hatanaka generated the anti-UBL3 antibody. H.A., M.S., and K.T. wrote the paper. All authors discussed and commented on the manuscript.

## Additional information

**Competing interests:** The authors declare no competing interests.

