## [Peer Review File · Nature Communications]

REVIEWERS' COMMENTS:

Reviewer #2 (Remarks to the Author):

The authors describe the role of ubiquitin-like 3 (UBL3) in regulating protein sorting to sEVs. The authors observed 60% reduction of total protein levels in serum sEVs which are purified from UBL3-deficient mice compared to wild-type mice. They also identified around 370 proteins belonging to extracellular vesicular exosome to be modified by UBL3. Lastly they provided the evidence that Ras and oncogenic RasG12V mutant are posttranslationally modified by UBL3, which led to increased sorting of RasG12V to sEVs. Overall, this is a well performed study describing a novel role for UBL3 in the sorting of proteins to sEVs. It is significantly improved version with clear evidence for the functional role of UBL3 in sorting of proteins but not RNA to sEVs. The work is acceptable for publication in Nature Communication.

Reviewer #3 (Remarks to the Author):

I believe the authors have appropriately addressed the questions and concerns of the reviewers through revision of the manuscript and the addition of new data. I have no additional comments.

Reviewer #3's comments on Reviewer #1's previous concerns

Comments for authors

The author's added new data including the use of additional antibodies such as CD9, CD63 and Alix along with the production of antibodies to UBL3 to define the vesicle populations more carefully and the data supports the exosome nature of the EVs. The author's also addressed the issue of total RNA and showed that RNA targeting to exosomes was not mediated through UBL3. Other points such as the lack of protein modification in HEK cells expressing endogenous UBL3 was also addressed.

Reviewer #4 (Remarks to the Author):

The revised manuscript from Ageta et. al. is much improved from the version I had seen. The experimental design has been changed substantively and the experiments seem to speak to the UBL3 modification more directly. The one thing that really seemed to puzzle me, and that wasn't clearly explained was why the Leu117 is so important for UBL3 modification of proteins. After all, the UBL3delta1 (CCVI) mutant goes to the membrane like WT-UBL3 but does not modify. This seems quite key to the understanding of UBL3 biology.

I recommend publication of the manuscript in Nature Communications.

REVIEWERS' COMMENTS:

Reviewer #2 (Remarks to the Author):

The authors describe the role of ubiquitin-like 3 (UBL3) in regulating protein sorting to sEVs. The authors observed 60% reduction of total protein levels in serum sEVs which are purified from UBL3-deficient mice compared to wild-type mice. They also identified around 370 proteins belonging to extracellular vesicular exosome to be modified by UBL3. Lastly they provided the evidence that Ras and oncogenic RasG12V mutant are posttranslationally modified by UBL3, which led to increased sorting of RasG12V to sEVs Overall, this is a well performed study describing a novel role for UBL3 in the sorting of proteins to sEVs. It is significantly improved version with clear evidence for the functional role of UBL3 in sorting of proteins but not RNA to sEVs. The work is acceptable for publication in Nature Communication.

Our reply

We thank Reviewer #2 for his/her overall agreement with the significance of our work and publication in Nature Communications.

Reviewer #3 (Remarks to the Author):

I believe the authors have appropriately addressed the questions and concerns of the reviewers through revision of the manuscript and the addition of new data. I have no additional comments.

Our reply

We appreciate Reviewer #3's assessment of addressing the questions and concerns and acknowledgement of the novelty and data quality of our work.

Reviewer #3's comments on Reviewer #1's previous concerns Comments for authors

The author's added new data including the use of additional antibodies such as CD9, CD63 and Alix along with the production of antibodies to UBL3 to define the vesicle populations more carefully and the data supports the exosome nature of the EVs. The author's also addressed the issue of total RNA and showed that RNA targeting to exosomes was not mediated through UBL3. Other points such as the lack of protein modification in HEK cells expressing endogenous UBL3 was also addressed.

Our reply

We truly thank Reviewer #3 again for efforts and time for checking all our comments to Reviewer #1 instead of Reviewer #1.

Reviewer #4 (Remarks to the Author):

The revised manuscript from Ageta et. al. is much improved from the version I had seen. The experimental design has been changed substantively and the experiments seem to speak to the UBL3 modification more directly.

Our reply

First of all, we thank Reviewer #4 for his/her overall agreement with the significance of our work.

The one thing that really seemed to puzzle me, and that wasn't clearly explained was why the Leu117 is so important for UBL3 modification of proteins. After all, the UBL3delta1 (CCVI) mutant goes to the membrane like WT-UBL3 but does not modify. This seems quite key to the understanding of UBL3 biology.

Our reply

We certainly agree with the reviewer # 4 that this point is extremely interesting. In fact, we have a preliminary data regarding specificity of Leu117. With reference to other CAAX proteins (Roberts et al., JBC 2008), we prepared several mutants in which Leu117 was replaced by T (Threonine), F (Phenylalanine), or M (Methionine). We also prepared mutants in which Leu117 was replaced by E (Glutamic Acid) as a negatively charged amino acid or R (Arginine) as a positively charged amino acid (please see the Right Figure a).

We tested UBL3 modification and membrane localization using these mutants. We expressed these mutants in MDA-MB-231 cells and purified UBL3 proteins by immunoprecipitation, followed by western blotting with UBL3 antiserum. For the detection of UBL3 modification, IP products were boiled without 2-mercaptoethanol (β ME-). A portion of the samples was treated with 2-mercaptoethanol (β ME+). As a result, we observed that UBL3 modification and membrane localization were retained and indistinguishable with wildtype either in L117T, L117F, L117M, or L117E. By contrast, UBL3 modification was attenuated in L117R. This result indicated that UBL3 modification occurs even if Leu117 is replaced by other amino acids such as Thr, Phe, Met or Glu. However, UBL3 modification was attenuated by substitution to Arg. Interestingly, UBL3L117R was found to have lost the membrane localization (please see the Right Figures b and c), indicating that membrane localization of UBL3 is important for the

UBL3 modification. Similar phenomenon was observed using UBL3C114A mutant in MDA-MB-231 (See Fig. 1d-e in our current study). These results indicate that membrane localization of UBL3 is necessary for the full activation of UBL3 modification.

Meanwhile, UBL3 Δ 1, which lacked only one carboxyl-terminal amino acid (Leu117), was found to have lost the UBL3 modification (Fig. 1g in the current study). However, UBL3 Δ 1 was retained in the membrane fraction (Fig. 1h in the current study). These results indicate that membrane localization is not sufficient for UBL3 modification.

Summarizing the above results, at the present time, although the function of Leu117 is not completely resolved, it is clear that it is important for UBL3 modification. Although Leu is not exchangeable with specific amino acid like Arg, it could be replaceable with other amino acids like Thr, Phe, Met or Glu.

There is CAAX motif for the membrane localization in the C-terminal of UBL3. It is also possible that there is a motif important for UBL3 modification. By performing other amino acid replacement, the motif necessary for UBL3 modification may be found. Furthermore, by searching for enzymes that recognize this motif sequence, an enzyme group related to the UBL3 modification can be found. Regarding the functional role of Leu117, we would like to address as a next project.

We agree with Reviewer #4's opinion. We also thought that Leu117 have an important role for UBL3 biology. Thus we added the following sentences to the discussion.

From the result of UBL3 Δ 1 mutant (Fig. 1g), the membrane localization of UBL3 alone is not sufficient for the UBL3 modification. Thus, in addition to CAAX motif for the membrane localization in the C-terminal of UBL3, it is possible that important amino acid sequences for the UBL3 modification in the C-terminal region exist. In the future UBL3 study, it is important to analyze molecules that recognize the C-terminal sequence for UBL3 modification.

We appreciate Reviewer #4 for giving us an opportunity to present our opinion in the discussion.

I recommend publication of the manuscript in Nature Communications.

Our reply

We thank again Reviewer #4 for his/her overall agreement with the significance of our work and recommendation in publication in Nature Communications.

Finally, we would like to thank all the reviewers' constructive comments and hope that we replied adequately to the issues.

We also hope that we are satisfactorily respond to comments, the revision and reply are suitable and our manuscript is now ready to publication in the current format.

Sincerely yours,

Kunihiro Tsuchida, M.D., Ph.D
Professor
Division for Therapies Against Intractable Diseases,
Institute for Comprehensive Medical Science (ICMS),
Fujita Health University
Toyoake, Aichi 470-1192, Japan

Mitsutoshi Setou, M.D., Ph.D
Professor
Department of Cellular and Molecular Anatomy,
Hamamatsu University School of Medicine,
1-20-1 Handayama, Higashi-ku, Hamamatsu, Shizuoka 431-3192, Japan